# Optical Properties of Red-Emitting Rb_2_Bi(PO_4_)(MoO_4_):Eu^3+^ Powders and Ceramics with High Quantum Efficiency for White LEDs

**DOI:** 10.3390/ma12193275

**Published:** 2019-10-08

**Authors:** Julija Grigorjevaite, Egle Ezerskyte, Agne Minderyte, Sandra Stanionyte, Remigijus Juskenas, Simas Sakirzanovas, Arturas Katelnikovas

**Affiliations:** 1Institute of Chemistry, Vilnius University, Naugarduko 24, LT-03225 Vilnius, Lithuania; julija.grigorjevaite@chf.vu.lt (J.G.); egle.ezerskyte@chf.stud.vu.lt (E.E.); agne.minderyte@chf.stud.vu.lt (A.M.); simas.sakirzanovas@chf.vu.lt (S.S.); 2Centre for Physical Sciences and Technology, Sauletekio Avenue 3, LT-10257 Vilnius, Lithuania; sandra.stanionyte@ftmc.lt (S.S.); remigijus.juskenas@ftmc.lt (R.J.)

**Keywords:** red phosphor, thermal quenching, quantum efficiency, luminescent ceramics, luminous efficacy, colour coordinates

## Abstract

There are several key requirements that a very good LED phosphor should meet, i.e., strong absorption, high quantum efficiency, high colour purity, and high luminescence quenching temperature. The reported Rb_2_Bi(PO_4_)(MoO_4_):Eu^3+^ phosphors have all these properties. The Rb_2_Bi(PO_4_)(MoO_4_):Eu^3+^ phosphors emit bright red light if excited with near-UV radiation. The calculated colour coordinates show good stability in the 77–500 K temperature range. Moreover, sample doped with 50% Eu^3+^ possesses quantum efficiency close to unity. Besides the powder samples, ceramic disks of Rb_2_Eu(PO_4_)(MoO_4_) specimen were also prepared, and the red light sources from these disks in combination with near-UV emitting LED were fabricated. The obtained results indicated that ceramic disks efficiently absorb the emission of 375 and 400 nm LED and could be applied as a red component in phosphor-converted white LEDs.

## 1. Introduction

In 1991 Nakamura invented the first efficient blue LED [1]. This invention has started the revolution in lighting industry. Besides the conventional incandescent and fluorescent lamps, the new alternative appeared to produce white light. The very first solid state light (SSL) sources used to combine the emission form a blue-emitting InGaN semiconductor chip with a yellow luminescence from a Y_3_Al_5_O_12_:Ce^3+^ phosphor, commonly known as YAG:Ce. Unfortunately, such combination of the blue and yellow emissions yielded bluish white light. Such light sources also possessed high correlated colour temperatures (CCT) and low colour rendering indices (CRI) due to lack of intensity in the red area [2]. One of the ways to solve this problem is to shift the Ce^3+^ emission in garnet type materials to the red spectral region. Based on this concept, some novel garnet phosphors emitting in the orange-red region were recently developed, for instance, Lu_2_CaMg_2_(Si,Ge)_3_O_12_:Ce^3+^ [2], Mg_3_Y_2_(Si,Ge)_3_O_12_:Ce^3+^ [3], Y_3_Mg_2_AlSi_2_O_12_:Ce^3+^ [4], and some others [5]. The other option to compensate the lack of intensity in the red area is to use the extra red phosphor. This research, in turn, resulted in several very efficient and stable orange/red-emitting nitride and oxynitride phosphors activated with Eu^2+^ ions: Ca_15_Si_20_O_10_N_30_:Eu^2+^ [6], (Ca,Sr)AlSiN_3_:Eu^2+^ [7], (Ba,Sr,Ca)_2_Si_5_N_8_:Eu^2+^ [8], Sr[LiAl_3_N_4_]:Eu^2+^ [9], and many others [10]. On the other hand, the synthesis of nitride/oxynitride phosphors usually requires protective gas atmosphere due to high reactivity of reagents, complicated equipment, high annealing temperatures, and, therefore, is very costly. Moreover, because of the rather broad emission band of Eu^2+^ ions, some of the emission lies in the deep red region (λ_em_ > 650 nm), where the human eye sensitivity is very low. According to Zukauskas et al., the emission at wavelengths higher than 650 nm can be considered as waste, because it reduces the luminous efficacy of the light source [11]. In this sense, the phosphor with narrow emission band or line in the red spectral region would be much superior.

With regular improvement of LEDs, the efficiency of near-UV emitting LEDs also was recently considerably improved and this has opened the new way for generating white light. The near-UV emitting LED can be combined with orange/red, green, and blue-emitting phosphors to yield white light source. This concept is somewhat superior if compared with blue LED driven light sources because more phosphors can be excited with near-UV radiation if compared to the blue LED. The efficient blue (BaMgAl_10_O_17_:Eu^2+^, BAM) [12] and green (Ba_2_SiO_4_:Eu^2+^, (Ca,Sr,Ba)Si_2_O_2_N_2_:Eu^2+^) [13,14] phosphors are already developed and are commercially available. Therefore, the main issue is finding the new narrow band or line emitting red phosphors with high efficiency, thermal stability and low production costs. In such research, the Eu^3+^ doped phosphors gained a lot of attention, especially the ones based on molybdate and tungstate host matrices. In these matrices Eu^3+^ possess unusually strong absorption due to admixing of charge transfer (CT) and Eu^3+^ excited states [15]. Recently, some efficient Eu^3+^ doped red-emitting molybdate phosphors were reported (LiEu(MoO_4_)_2_ [16], Y_2_Mo_4_O_15_:Eu^3+^ [15], K_2_Bi(PO_4_)(MoO_4_):Eu^3+^ [17], Tb_2_Mo_3_O_12_:Eu^3+^ [18]) and the search for novel materials in this class is still ongoing [19]. Besides, recently interest in Mn^4+^ doped inorganic materials is also gaining lots of attention as an alternative for narrow band red-emitting phosphors. Elimination of lanthanide ions from the luminescent material would of course significantly reduce the phosphor price, what is a great benefit for industrial application. The research in this area now mostly concentrates on oxide and fluoride host matrices with the aim to shift Mn^4+^ emission from deep red to red region [20,21,22].

In this research, the Rb_2_Bi(PO_4_)(MoO_4_):Eu^3+^ luminescent materials were synthesized by conventional solid-state reaction method at fairly low temperature. The optical properties of synthesized phosphors were investigated as a function of excitation wavelength, temperature, and Eu^3+^ concentration. The given data will include reflection, emission and excitation spectra at room temperature and in 77–500 K interval; photoluminescence (PL) lifetimes, quantum efficiencies, luminous efficacies (LE) and CIE 1931 colour coordinates.

## 2. Materials and Methods 

A series of Rb_2_Bi(PO_4_)(MoO_4_):Eu^3+^ samples of (where Eu^3+^ concentration was 0%, 1%, 5%, 10%, 25%, 50%, 75%, and 100% with respect to Bi^3+^) was synthesized by solid-state reaction method. The reagents, namely, MoO_3_ (99+%), Rb_2_CO_3_ (99+%) and Bi_2_O_3_ (99.9%) were purchased from Acros Organics (Geel, Belgium); Eu_2_O_3_ (99.99%) was purchased from Tailorlux (Münster, Germany), and NH_4_H_2_PO_4_ (99%) was purchased from Reachem Slovakia, (Bratislava, Slovakia). These reagents were weighed and mixed in stoichiometric amounts (accurate weight of reagents are given in Appendix A). The powders were homogenized in an agate mortar with small amount of acetone as a grinding media. The mixture of reagents was poured to the porcelain crucible and sintered at 600 °C for 12 h in air three times with intermediate grinding of the product.

0.36, 0.53, and 0.80 mm thick ceramic disks were prepared by applying 30 kN force (ø 8 mm disk) on the Rb_2_Eu(PO_4_)(MoO_4_) phosphor powder for 3 min and the obtained pellets were additionally annealed in air at 600 °C for 4 h.

The powder XRD patterns were collected in the range of 5° ≤ 2θ ≤ 80° employing Cu Kα radiation from 9 kW rotating anode X-ray tube on a Rigaku SmartLab diffractometer (Rigaku, Tokyo, Japan) working in Bragg-Brentano focusing geometry. The step width was 0.01° and scan speed was 1°/min. Scintillation detector SC-70 (Rigaku, Tokyo, Japan) together with bent graphite monochromator (Rigaku, Tokyo, Japan) for diffracted beam was used for the measurements.

SEM images of the prepared phosphors and ceramics were taken by a field-emission scanning electron microscope Hitachi SU-70 (Hitachi, Tokyo, Japan).

IR measurements were performed on a Bruker Alpha ATR FTIR spectrometer (Bruker, Ettlingen, Germany). Data were recorded in the range from 3000 to 400 cm^−1^. The spectral resolution was 4 cm^−1^.

Edinburgh Instruments FLS980 spectrometer (Edinburgh Instruments, Livingston, UK) was used to measure reflection spectra. The spectrometer possessed a 450 W ozone free xenon arc lamp, a cooled (−20 °C) SPC photomultiplier (Hamamatsu R928P, Hamamatsu Photonics K.K., Hamamatsu, Japan) and Teflon coated integration sphere. The reflectance standard was BaSO_4_.

Edinburgh Instruments FLS980 spectrometer was used to record excitation and emission spectra of the synthesized samples. This spectrometer was equipped with double grating Czerny-Turner excitation and emission monochromators, a 450 W ozone-free xenon arc lamp, and cooled (−20 °C) SPC photomultiplier (Hamamatsu R928P). The recorded PL emission spectra were corrected for instrument response by a correction file provided by Edinburgh Instruments (Livingston, UK). A reference detector (Edinburgh Instruments, Livingston, UK) was used to correct the excitation spectra.

For thermal quenching (TQ) measurements a cryostat “MicrostatN” (Oxford Instruments, Abingdon, UK) was used with above-described spectrometer. The sample was cooled by liquid nitrogen. The temperature-dependent emission spectra were measured at 77 K and at 100–500 K in 50 K intervals. The stabilization time for each temperature was 120 s and temperature tolerance was ±5 K. Dried nitrogen was flushed over the cryostat window during the measurements in order to prevent water condensation at low temperatures.

μ-flash lamp was used as an excitation source for PL decay investigation. Samples were excited at 265, 393.5, and 464.5 nm while emission was monitored at 615 nm. The pulse repetition rate was 25 Hz.

Quantum efficiency (QE) values were obtained by measuring emission spectrum of BaSO_4_ sample (99% Sigma-Aldrich) in integration sphere coated with Teflon. Samples were excited at 260, 393.5, and 464.5 nm. The step width and integration time was 0.5 nm and 0.4 s, respectively. Measurements were repeated three times for each phosphor sample. The QE values were calculated from equation 1 [17]:(1)$$QE=\frac{\int I_{em,\;sample}-\int I_{em,\;BaSO_{4}}}{\int I_{ref,\;BaSO_{4}}-\int I_{ref,\;sample}}\times 100\%=\frac{N_{em}}{N_{abs}}\times 100\%$$

Here $\int I_{em,\;sample}\;$ is integrated emission intensity of the phosphor sample and $\int I_{em,\;BaSO_{4}}$ is integrated emission intensity of BaSO_4_. $\int I_{ref,\;sample}\;$ and $\int I_{ref,\;BaSO_{4}}$ is integrated reflectance of the phosphor sample and BaSO_4_, respectively. *N_em_* is the number of emitted photons and *N_abs_*—absorbed photons.

All measurements were performed at room temperature and ambient pressure in air unless mentioned otherwise.

## 3. Results and Discussion

The Rb_2_Bi(PO_4_)(MoO_4_) compound (a = 7.0671 Å, b = 12.5150 Å, c = 20.441 Å) [23] is isostructural with its potassium counterpart (K_2_Bi(PO_4_)(MoO_4_)) (a = 7.0296 Å, b = 12.4845 Å, c = 19.7146 Å) [17] and adopts a body centered orthorhombic Bravais lattice with the space group of *Ibca* (#73) (Z = 8). The crystal structure of Rb_2_Bi(PO_4_)(MoO_4_) compound possesses a layered structure, which is built from [Bi(PO_4_)(MoO_4_)]^2-^ layers, which and are interconnected by Rb^+^ cations similar as in K_2_Bi(PO_4_)(MoO_4_) compound reported by Zatovsky et al. [24]. The graphical representation of Rb_2_Bi(PO_4_)(MoO_4_) unit cell is given in Appendix A.

It turned out that three annealing steps were necessary in order to obtain single phase Rb_2_Bi(PO_4_)(MoO_4_):Eu^3+^ compounds. The powder XRD patterns of Rb_2_Eu(PO_4_)(MoO_4_) specimen after each annealing step are given in Appendix A. It is obvious that the peaks belonging to the phases other than Rb_2_Bi(PO_4_)(MoO_4_) disappear only after the third heat treatment. Moreover, Appendix A also demonstrates that single phase compounds are obtained despite the increasing Eu^3+^ amount in the synthesized materials. The representative powder XRD pattern of Rb_2_Eu(PO_4_)(MoO_4_) compound is given in Figure 1. All the peaks match well with the reference pattern of K_2_Bi(PO_4_)(MoO_4_). The only difference in these XRD patterns is the slight shift of peaks for Rb_2_Eu(PO_4_)(MoO_4_) compound to the lower 2θ values. On the other hand, this can be expected since the ionic radius of potassium ions is slightly smaller than for rubidium ions, 1.51 and 1.61 Å for CN = 8, respectively [25].

The morphological features of all prepared materials were investigated by taking SEM images. No differences neither in particle size nor shape were observed with the increase of Eu^3+^ concentration in synthesized compounds. The representative SEM images of Rb_2_Bi(PO_4_)(MoO_4_):50%Eu^3+^ specimen are given in Figure 2a,b. The figure demonstrates that powder consists of agglomerates, which are formed from smaller crystallites with broad size range. The shape of these crystallites is mostly rod-like. Figure 2c depicts the SEM image of the surface of Rb_2_Eu(PO_4_)(MoO_4_) ceramic disk. The ceramic disk surface is formed from crystallites of various sizes. These crystallites are well grown together and form an even surface of the ceramic disk.

The IR spectra of undoped Rb_2_Bi(PO_4_)(MoO_4_) sample and specimens doped with 25%, 50%, 75% and 100% Eu^3+^ are shown in Appendix A. The given IR spectra contain several absorption bands in the range of 400–1100 cm^−1^. The three sharp absorption bands in the range of 650–450 cm^−1^ can be assigned to the PO_4_ bending vibrations. Strong absorption band at 900–700 cm^−1^ originates from the Mo-O stretching vibrations in MoO_4_ tetrahedrons, whereas strong absorption bands at 950 and 1050 cm^−1^ can be assigned to symmetric and asymmetric vibrations of PO_4_ tetrahedrons, respectively [24].

Figure 3a depicts excitation spectra of Rb_2_Bi(PO_4_)(MoO_4_) samples doped with 1% and 100% Eu^3+^ ions. The spectra consist of a broad band in the range of 250–290 nm and several sets of lines in the range of 290–600 nm. The former can be assigned to the charge transfer transition from O^2−^ ions to Eu^3+^ ions, whereas the latter is assigned to the intraconfigurational transitions within Eu^3+^
*f* orbital ([Xe]4f^6^ → [Xe]4f^6^). Each set of lines in excitation spectra can be attributed to the specific transition, i.e., ^7^F_0_ → ^5^F_J_ (~298 nm), ^7^F_0_ → ^5^H_J_ (~319 nm), ^7^F_0_ → ^5^D_4_ (~362 nm), ^7^F_0_ → ^5^L_7,8_; ^5^G_J_ (~380 nm), ^7^F_0_ → ^5^L_6_ (~394 nm), ^7^F_1_ → ^5^D_3_ (~418 nm), ^7^F_0_ → ^5^D_2_ (~467 nm), ^7^F_0_ → ^5^D_1_ (~527 nm), ^7^F_1_ → ^5^D_1_ (~534 nm), ^7^F_0_ → ^5^D_0_ (~580 nm), and ^7^F_1_ → ^5^D_0_ (~590 nm) [26,27]. The lines in excitation spectra lying in the range of 360–400 nm are of great interest, since they overlap well with the emission spectra of the most efficient near-UV LEDs [15].

Emission spectra of Rb_2_Bi(PO_4_)(MoO_4_) specimens doped with 1% and 100% Eu^3+^ ions are given in Figure 3b. These spectra contain five sets of emission lines, which can be assigned to Eu^3+^ electronic transitions from the lowest excited ^5^D_0_ level to ^7^F_0_ (~580 nm), ^7^F_1_ (~594 nm), ^7^F_2_ (~615 nm), ^7^F_3_ (~652 nm), and ^7^F_4_ (~702 nm) [27]. The weakest emission was observed for the ^5^D_0_ → ^7^F_0_ transition since the *J* = 0 ↔ *J*′ = 0 transitions are always forbidden [28,29,30]. This transition can be observed because the local site of Eu^3+^ ions possesses C_2_ symmetry. The strongest emission of the prepared samples was observed for the electric dipole (ED) ^5^D_0_ → ^7^F_2_ transition at around 615 nm. The emission lines originating from the magnetic dipole (MD) ^5^D_0_ → ^7^F_1_ transition was more than twice less intensive. This observation indicates that Eu^3+^ ions occupy non-centrosymmetric lattice site in the Rb_2_Bi(PO_4_)(MoO_4_) host matrix. On the other hand, the intensity difference between MD and ED transitions is rather low if compared to some other molybdate host matrices doped with Eu^3+^ ions, for instance, Li_3_Ba_2_La_3_(MoO_4_):Eu^3+^ [31], Tb_2_Mo_3_O_12_:Eu^3+^ [18], La_2_MoO_6_:Eu^3+^ [32], LiEuMo_2_O_8_ [33], La_2_Mo_2_O_9_:Eu^3+^ [34] and so on. The rather small difference between ED and MD transitions of Eu^3+^ ions in Rb_2_Bi(PO_4_)(MoO_4_) host matrix can be assigned to relatively symmetric dodecahedral BiO_8_ site, which is occupied by Eu^3+^. Another interesting feature of the Rb_2_Bi(PO_4_)(MoO_4_):Eu^3+^ emission spectra is very intensive and finely split ^5^D_0_ → ^7^F_4_ transition. The intensity of ^5^D_0_ → ^7^F_4_ transition usually is very low in Eu^3+^ doped phosphors and there are very few examples reported in the literature where this transition dominates the emission spectra. Some of these matrices are garnet structure compounds: Y_3_(Al,Ga)_5_O_12_:Eu^3+^ [35]; orthophosphates: (Lu,Y,Gd,La)PO_4_:Eu^3+^ [35], BiPO_4_:Eu^3+^ [36]; borates: GdB_5_O_9_:Eu^3+^ [37]; molybdates: Eu_2_Mo_4_O_15_ [38], silicates: Na_2_ZnSiO_4_:Eu^3+^ [39], Ca_2_Ga_2_SiO_7_:Eu^3+^ [40]; aluminates: GdSr_2_AlO_5_:Eu^3+^ [41]; niobates: K_2_LaNb_5_O_15_:Eu^3+^ [42]; and even exotic uranyl phosphates: (Y,Eu,La)(UO_2_)_3_(PO_4_)_2_O(OH)·6H_2_O [43]. The inset in Figure 3b shows integrated emission intensity as a function of Eu^3+^ concentration. It can be concluded, that the strongest emission is obtained for sample doped with 50% Eu^3+^.

The reflection spectra of undoped Rb_2_Bi(PO_4_)(MoO_4_) and its fully Eu^3+^ substituted counterpart Rb_2_Eu(PO_4_)(MoO_4_) are given in Figure 3c. The powder colour of undoped Rb_2_Bi(PO_4_)(MoO_4_) was yellowish what is in good agreement with reflection spectra. The host material slightly absorbs the violet-blue radiation, hence, the yellowish tint of the powder. The fully Eu^3+^ substituted compound, in turn, was white. There are also several sets of absorption lines originating from the intraconfigurational transitions of Eu^3+^ ions that very well match the lines observed in excitation spectra (Figure 3b). The simplified energy level diagram of Eu^3+^ ions is given in Figure 3e with the most intensive transitions marked.

Figure 3d displays PL decay curves of Rb_2_Bi(PO_4_)(MoO_4_) specimens doped with 1% and 100% Eu^3+^. Samples were excited at 393.5 nm whereas the emission was monitored at 615 nm. Both PL decay curves are linear, indicating that there is only one mechanism of the excited ^5^D_0_ level depopulation. Besides, both PL decay curves are also nearly the same indicating similar PL lifetime values. This turned out to be true when the single exponential decay function was used to extract the PL lifetime values from the experimental data:(2)$$I(t)=A+Be^{-\frac{t}{\tau }}$$

Here *I(t)* is PL intensity at time *t*, *A* is background, *B* is constant, and *τ* is PL lifetime. The fitting results are given in Appendix A. The PL lifetime values increased from 1827 μs (for 1% Eu^3+^) to 2044 μs (for 50% Eu^3+^) and then again decreased to 1932 μs (for 100% Eu^3+^). This goes hand in hand with the emission integral data, which showed highest intensity for 50% Eu^3+^ doped sample. Therefore, it can be concluded that concentration quenching in this host matrix starts when Eu^3+^ concentration exceeds 50%.

The temperature-dependent excitation spectra (λ_em_ = 615 nm) of Rb_2_Eu(PO_4_)(MoO_4_) sample recorded at 77 and 500 K are given in Figure 4a. These spectra show some considerable differences. First of all, the charge transfer band becomes much broader at elevated temperatures, meaning that electron from O^2−^ ions is transferred to Eu^3+^ ions much easier at high temperatures. Besides that, some lines in excitation spectra also disappear at low temperatures. For instance, lines that originate from thermally populated ^7^F_1_ level are not observed in excitation spectra recorded at 77 K. These include ^7^F_1_ → ^5^D_3_, ^7^F_1_ → ^5^D_1_ and ^7^F_1_ → ^5^D_0_ transitions occurring at around 415, 534 and 590 nm, respectively. The absence of excitation lines from ^7^F_1_ level indicates that thermal population of ^7^F_1_ level at low temperatures is very limited.

The normalized emission spectra (λ_ex_ = 393.5 nm) of Rb_2_Eu(PO_4_)(MoO_4_) sample recorded at 77 and 500 K are given in Figure 4b. These spectra consist of the same five sets of emission lines as spectra recorded at room temperature (see Figure 3b). The only difference between emission spectra recorded at 77 and 500 K is increased broadening of the lines at higher temperatures. This phenomenon can be attributed to the increased lattice vibrations at elevated temperatures which leads to distortion of Eu^3+^ local surrounding. This results in Eu^3+^ emission at slightly different wavelengths; hence, the broader emission lines.

The emission stability of the phosphor at elevated temperatures is an important factor determining the phosphors suitability for practical applications. The high power near-UV or blue LEDs can heat up to temperatures as high as 150 °C; therefore, the phosphor considered for such application should emit at this temperature efficiently. Besides, the temperature-dependent integrated emission spectra can also be used to determine the thermal quenching *TQ_1/2_* (this value shows the temperature at which phosphor loses half of its efficiency) and thermal quenching activation energy *Ea* values. This is usually done by applying Fermi-Dirac model [18,44]:(3)$$\frac{I(T)}{I_{0}}=\frac{1}{1+Be^{-Ea/kT}}$$

Here *I(T)* and *I_0_* are the temperature-dependent emission integral and the highest value of emission integral, respectively. *B* is quenching frequency factor. *Ea* is thermal quenching activation energy. *T* is absolute temperature and *k* is Boltzmann constant (8.617342·10^−5^ eV/K) [45]. The *TQ_1/2_* values, in turn, can be derived by slight changes of equation 3 [42]:(4)$$TQ_{1/2}=\frac{-Ea}{k\times \ln(1/B)}$$

The calculated *TQ_1/2_* and *Ea* values for Rb_2_Eu(PO_4_)(MoO_4_) specimen are 716 K ±46 K and 0.1391 eV ±0.0142 eV, respectively (see Figure 4c). These results demonstrate that Rb_2_Eu(PO_4_)(MoO_4_) sample would lose half efficiency at temperatures well above of the typical high power LED; therefore, it could be applied as a red component in near-UV based white LEDs. Besides, the *TQ_1/2_* values of Rb_2_Eu(PO_4_)(MoO_4_) sample are also higher than those reported for its potassium counterpart K_2_Eu(PO_4_)(MoO_4_) reported earlier in the literature (*TQ_1/2_* = 578 K ± 11 K) [17]. In this sense, the Rb_2_Eu(PO_4_)(MoO_4_) is superior if compared to K_2_Eu(PO_4_)(MoO_4_).

The temperature-dependent PL decay curves were also recorded for Rb_2_Bi(PO_4_)(MoO_4_) samples doped with 1%, 50%, and 100% Eu^3+^. The PL decay profile was single exponential and the PL lifetime values were calculated employing equation 2. The obtained temperature-dependent PL lifetime values for those three specimens are depicted in Figure 4d. The exact PL lifetime values together with standard deviations are summarized in Appendix A. It is interesting to note that PL lifetime values of samples doped with 1% and 50% Eu^3+^ decreases with increasing temperature. However, the PL lifetime values of specimen doped with 100% Eu^3+^ increases up to 400 K and only then start to decrease. This shows, that the internal quantum efficiency of this phosphor is the highest at 400 K.

Another important specification of the phosphor considered for application in white LEDs is the colour purity. This can be evaluated by calculating the colour coordinates from the respective emission spectra. The fragments of CIE 1931 colour space diagram together with Eu^3+^ concentration-dependent colour coordinates at room temperature and temperature-dependent colour coordinates of 1%, 50%, and 100% Eu^3+^ doped samples are depicted in Figure 5a–d, respectively. All the colour coordinates are located directly on the edge of the CIE 1931 colour space diagram showing high colour purity of all synthesized materials. It is also evident that colour coordinates for samples doped with higher Eu^3+^ concentration shift downward the edge of the CIE 1931 colour space diagram. This shows that heavier doping leads to slightly more red emission of the samples. It turned out, that temperature also influences the colour coordinates slightly. They shift upwards the edge of the CIE 1931 colour space diagram indicating that phosphors give slightly more intensity in the orange spectral region at increased temperatures. On the other hand, the shift is relatively small and the colour coordinates can be considered stable within the 77–500 K range. The exact colour coordinate values in CIE 1931 colour space as a function of Eu^3+^ concentration and temperature are tabulated in Appendix A and Appendix A, respectively. The colour coordinates of the prepared samples are virtually the same regardless of the excitation wavelength (265, 393.5 or 465 nm) as shown in Appendix A.

The brightness of emission of the prepared samples to human eye can be evaluated by calculating the luminous efficacy (LE) values from the emission spectra. These values were obtained by employing the following equation [46]:(5)$$LE(lm/W_{opt})=683(lm/W_{opt})\times \frac{\int I(\lambda )\;V(\lambda )\;d\lambda }{\int I(\lambda )\;d\lambda }$$

Here, *I(λ)* is emission spectrum of the sample and *V(λ)* is the human eye sensitivity curve. The human eye is the most sensitive to the 555 nm electromagnetic radiation; therefore, the highest possible luminous efficacy value of 683 lm/W_opt_ is obtained for monochromatic green light at 555 nm. The obtained LE values for the synthesized samples as a function of Eu^3+^ concentration and excitation wavelength (265, 393.5, and 465 nm) are tabulated in Appendix A whereas the temperature-dependent LE values for samples doped with 1%, 50%, and 100% Eu^3+^ are given in Appendix A. The LE values of all prepared samples were virtually the same regardless the Eu^3+^ concentration, excitation wavelength or temperature and varied around 200 lm/W_opt_. The luminous efficacy values of the synthesized phosphors are relatively high and are close to or even superior than those reported for well-known broad band emitting Eu^2+^ phosphors, for instance, CaS:Eu^2+^ (LE = 85 lm/W_opt_, λ_em_ = 650 nm), Sr_2_Si_5_N_8_:Eu^2+^ (LE = 240 lm/W_opt_, λ_em_ = 620 nm), CaAlSiN_3_:Eu^2+^ (LE = 150 lm/W_opt_, λ_em_ = 650 nm) [47]. On the other hand, the synthesized phosphors as was already mentioned show strong ^5^D_0_ → ^7^F_4_ emission at around 702 nm and this reduces the LE values since human eye is insensitive in this range. This leads to lower LE values if compared to other Eu^3+^ doped phosphors where ^5^D_0_ → ^7^F_4_ emission is not so strong, for example, Li_3_Ba_2_Eu_3_(MoO_4_)_8_ (LE = 312 lm/W_opt_, λ_em_ = 615.5 nm) [31] or Y_2_Mo_4_O_15_:75%Eu^3+^ (LE = 242 lm/W_opt_, λ_em_ = 613 nm) [15].

The quantum efficiencies (*QE*) of all prepared samples as a function of excitation wavelength and Eu^3+^ concentrations are depicted in Figure 6. The *QE* value for the 50% Eu^3+^ doped sample is close to unity under 393.5 nm excitation. Such high *QE* value is very favourable for practical application. Further increase of the Eu^3+^ concentration resulted in decreased *QE* values under the same excitation wavelength. Moreover, excitation at 265 and 465 nm has also resulted in somewhat lower *QE* values. This leads to a conclusion that the synthesized phosphors in practical application should be used with near-UV emitting LEDs.

Due to forbidden nature of the intraconfigurational Eu^3+^ [Xe]4f^6^ ↔ [Xe]4f^6^ transitions the absorption of Eu^3+^ ions in inorganic matrices are relatively low [48]. One of the ways to deal with this problem is using ceramics prepared from phosphor powder instead of phosphor powder itself. In ceramics the scattering of the incident excitation light is suppressed; therefore, the light can penetrate deeper in the phosphor and thus excite more dopant ions. In order to test this concept, we have prepared three ceramic disks with thicknesses of 0.36, 0.53, and 0.80 mm from Rb_2_Eu(PO_4_)(MoO_4_) sample. The quantum efficiency of this sample is 10% lower if compared to sample doped with 50% Eu^3+^. However, the powder of fully Eu^3+^ substituted phosphor possesses stronger absorption; thus, in our opinion, this trade of is very reasonable. The prepared ceramic disks were placed on three different LEDs, emitting at 375, 400, and 455 nm, and the emission spectra of the resulting light sources were measured. The obtained results are summarized in Figure 7. For comparison, the emission spectra of 375, 400, and 455 nm LEDs are given in Figure 7a–c, respectively. Figure 7d demonstrates that the absorption of electromagnetic radiation emitted by 375 nm LED increases with increasing thickness of the Rb_2_Eu(PO_4_)(MoO_4_) ceramic disc. The ceramic disc thickness of 0.36 mm is not sufficient to absorb all the light emitted by 375 nm LED; however, it is virtually all absorbed when the thickness of ceramic disk reached 0.80 mm. The Eu^3+ 7^F_0_ → ^5^L_7,8_; ^5^G_J_ and ^7^F_0_ → ^5^L_6_ absorption lines are clearly visible in the emission spectra of the obtained light source where 375 nm LED emits. Similar behaviour was observed for light sources prepared from 400 nm emitting LED and ceramic disks. The only difference is that 0.80 mm thickness ceramic disk is still not enough to absorb all the incident radiation; therefore, for complete absorption of 400 nm LED emission a thicker ceramic disk would be required. Here, Eu^3+^ absorption lines originating from Eu^3+ 7^F_0_ → ^5^L_6_ and ^7^F_1_ → ^5^D_3_ transitions are visible in emission spectra of 400 nm LED. The lowest absorption of ceramic disks was observed if 455 LED was used for excitation. This, however, is not surprising since in this spectral range only one absorption line (namely, ^7^F_0_ → ^5^D_2_) of Eu^3+^ ions is present. Moreover, this line is very narrow and even though nearly all LED emission at around 465 nm is absorbed, the rest of LED emission passes through the disk unabsorbed. The relatively low absorption of 375 and 465 nm LED emission by prepared ceramic disks is, of course, a disadvantage for the practical application. However, on the positive side, the unabsorbed LED emission remains available for exiting phosphors emitting in other spectral regions.

The colour coordinates and LE values of the obtained light sources were also calculated. The exact values are tabulated in Appendix A whereas the graphical representation is shown in Figure 8. The colour coordinates of the light sources obtained from Rb_2_Eu(PO_4_)(MoO_4_) ceramic discs and 375 nm emitting LED shift closer to the edge of CIE 1931 colour space diagram if ceramics thickness is increased (see Figure 8a). This shows high colour purity of the emitted red light. Such result also goes hand in hand with emission spectra depicted in Figure 7d, where emission from LED is completely absorbed by the thickest ceramic disk. Different results, however, were obtained when 400 nm emitting LED was used as an excitation source. The colour coordinates of these light sources lied in the purplish red spectral region of the CIE 1931 colour space diagram and shifted towards red region with increasing ceramics thickness (see Figure 8b). These results correlate well with emission spectra given in Figure 7e. The blend of Eu^3+^ emission in ceramic disks and unabsorbed emission from LED yields the purplish red emission colour. The fraction of LED emission decreases with increasing ceramics thickness; hence, the red shift of colour coordinate. It turned out that no red light source can be obtained by combining Rb_2_Eu(PO_4_)(MoO_4_) ceramic discs and 455 nm emitting LED. The absorption lines in this region by Eu^3+^ ions are just too narrow. Therefore, the major part of 455 nm LED emission passes through the ceramic disks unabsorbed regardless their thickness and all the colour coordinates of such light sources lie in the blue region of the CIE 1931 colour space diagram as shown in Figure 8c.

The highest LE values were obtained for light sources obtained from Rb_2_Eu(PO_4_)(MoO_4_) ceramic discs and 375 nm emitting LED. The LE values increased from 130 to 186 lm/W_opt_ with increasing ceramic thickness from 0.36 to 0.80 mm. This is associated with decreasing fraction of 375 nm LED emission for which human eye is extremely insensitive. Similar results were observed for the light sources obtained using 400 nm emitting LED. Here the LE values increased from 98 to 156 lm/W_opt_. The lower LE values, if compared to devices with 375 nm emitting LED, are obtained because a larger fraction of LED emission remains unabsorbed. Finally, the lowest luminous efficacy values were calculated for light sources using 455 nm emitting LED. In this case the LE values were only 55 and 73 lm/W_opt_ for devices with the thinnest and the thickest ceramic disks, respectively. This is due to emission of such light sources in the blue and red spectral regions were the human eye sensitivity is low.

## 4. Conclusions

Single phase Rb_2_Bi(PO_4_)(MoO_4_):Eu^3+^ phosphors with Eu^3+^ concentration ranging from 1% to 100% were synthesized by solid state reaction method at relatively low temperatures. All samples showed bright red luminescence if excited with near-UV radiation. The most intensive emission lines of Eu^3+^ ions were observed for ^5^D_0_ → ^7^F_2_ and ^5^D_0_ → ^7^F_4_ transitions at 615 and 702 nm, respectively. The calculated colour coordinates also indicated high colour purity of the emission regardless the Eu^3+^ concentration and temperature. Samples, excited at 393.5 nm possessed high quantum efficiency reaching almost 100% for 50% Eu^3+^ doped specimen. Moreover, these phosphors also showed high emission stability at elevated temperatures. The extrapolation of experimental results indicated that synthesized phosphors would lose half of efficiency only at 716 K what is well beyond the operating temperature of high power LED semiconductor chip. The calculated luminous efficacies were comparable or even superior to those reported for well-established Eu^2+^ doped phosphors. All these properties of herein reported Rb_2_Bi(PO_4_)(MoO_4_):Eu^3+^ phosphors makes them good candidates for practical application as a red component in a near-UV LED driven solid state light sources. However, it should also be noted that the most efficient samples possess high concentration of Eu^3+^ which makes these phosphors relatively expensive. Moreover, these phosphors could also be considered for application in luminescent security pigments due to unique emission spectra.

## Figures and Tables

**Figure 1 materials-12-03275-f001:**
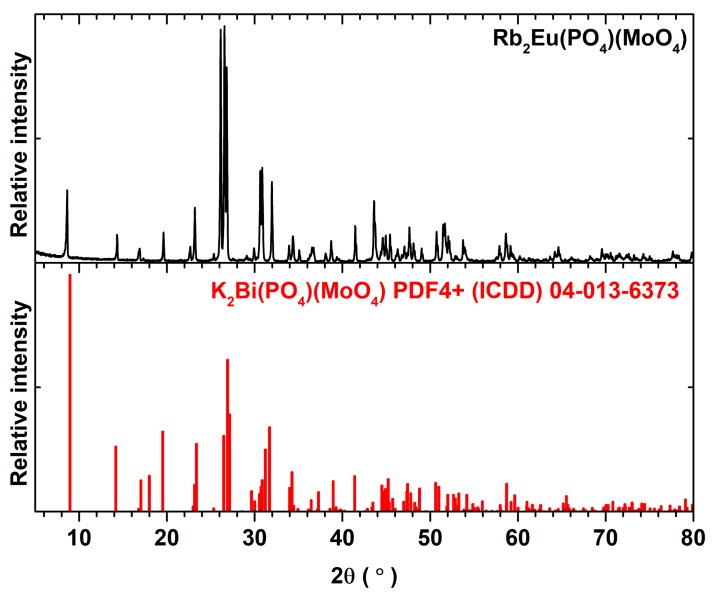
Powder XRD pattern of Rb_2_Eu(PO_4_)(MoO_4_) specimen together with reference pattern of K_2_Bi(PO_4_)MoO_4_) (PDF4+ (ICDD) 04-013-6373).

**Figure 2 materials-12-03275-f002:**
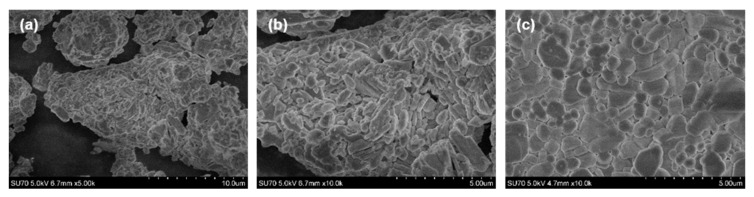
SEM images of Rb_2_Bi(PO_4_)(MoO_4_):50%Eu^3+^ powders (**a**,**b**) and ceramic disk of Rb_2_Eu(PO_4_)(MoO_4_) (**c**).

**Figure 3 materials-12-03275-f003:**
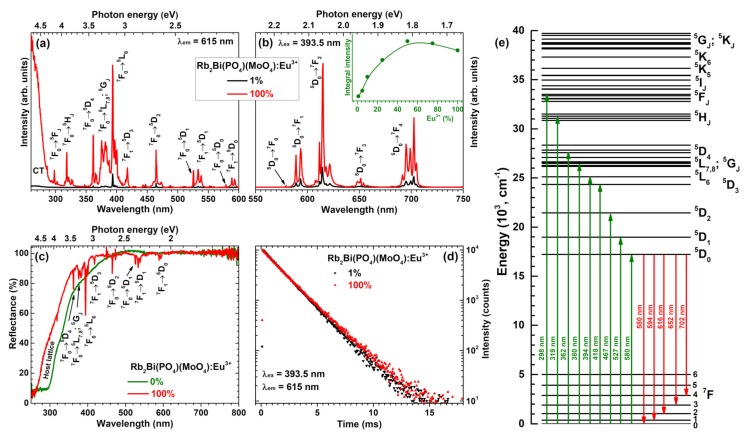
(**a**) Excitation (λ_em_ = 615 nm) and (**b**) emission (λ_ex_ = 393.5 nm) spectra of Rb_2_Bi(PO_4_)(MoO_4_):Eu^3+^; inset shows emission integral intensity as a function of Eu^3+^ concentration (λ_ex_ = 393.5 nm). (**c**) Reflection spectra of Rb_2_Bi(PO_4_)(MoO_4_) and Rb_2_Eu(PO_4_)(MoO_4_). (**d**) Photoluminescence decay (λ_ex_ = 393.5 nm, λ_em_ = 615 nm) curves of Rb_2_Bi(PO_4_)(MoO_4_) samples doped with 1% and 100% Eu^3+^. (**e**) Energy levels of Eu^3+^ ions with wavelengths of electronic transitions.

**Figure 4 materials-12-03275-f004:**
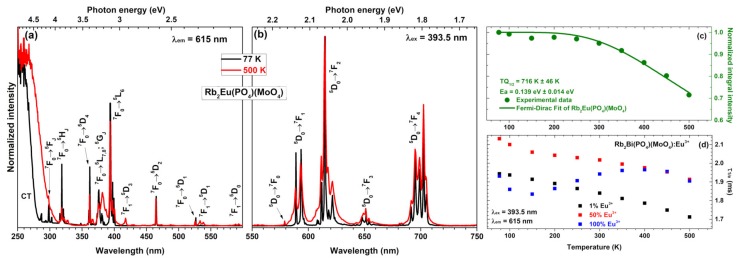
(**a**) Excitation (λ_em_ = 615 nm) and (**b**) emission (λ_ex_ = 393.5 nm) spectra of Rb_2_Eu(PO_4_)(MoO_4_) at 77 and 500 K temperature. (**c**) Calculation of TQ_1/2_ value for the sample doped with 100% Eu^3+^ from normalized emission integral intensity. (**d**) Temperature-dependent photoluminescence (PL) lifetimes (τ_1/e_) of Rb_2_Bi(PO_4_)(MoO_4_) doped with 1%, 50%, and 100% Eu^3+^.

**Figure 5 materials-12-03275-f005:**
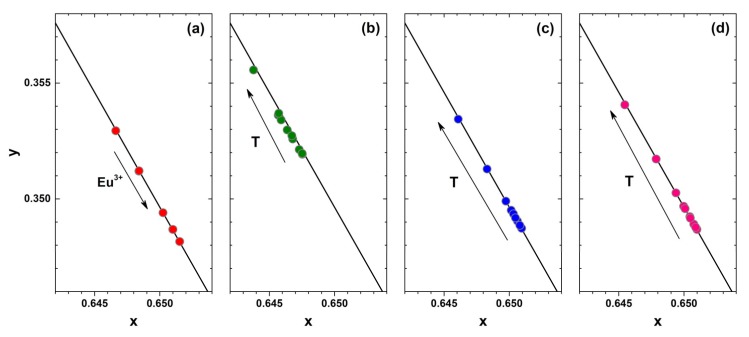
Fragments of the CIE 1931 colour diagrams with the colour points of (**a**) Rb_2_Bi(PO_4_)(MoO_4_):Eu^3+^ as a function of Eu^3+^ concentration and as a function of temperature of (**b**) 1%, (**c**) 50%, and (**d**) 100% Eu^3+^ doped samples. All samples were excited at 393.5 nm.

**Figure 6 materials-12-03275-f006:**
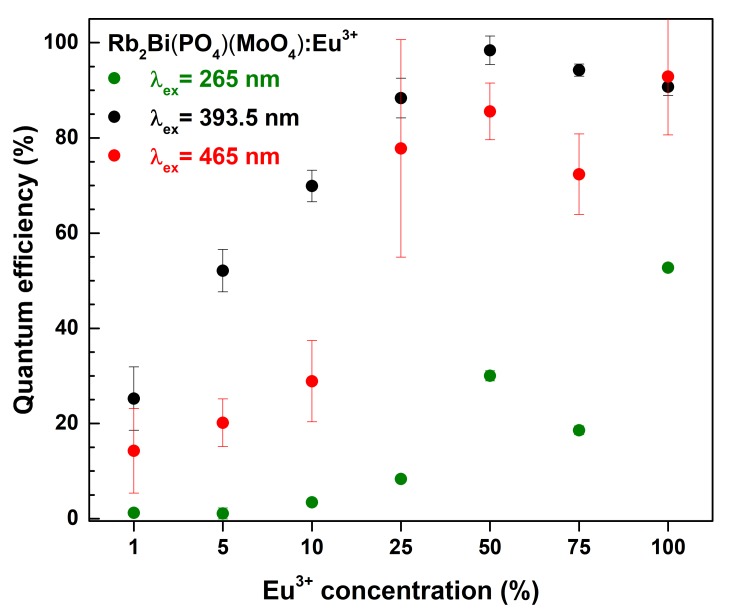
Quantum efficiencies of Rb_2_Bi(PO_4_)(MoO_4_):Eu^3+^ phosphors as a function of Eu^3+^ concentration and excitation wavelength.

**Figure 7 materials-12-03275-f007:**
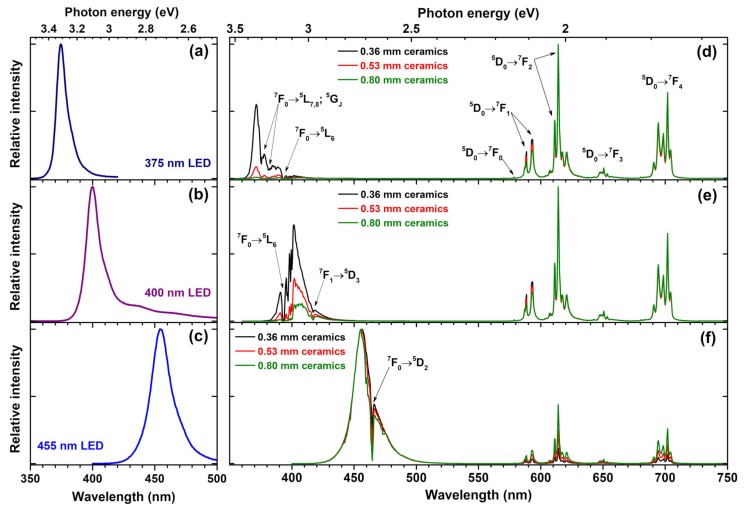
Emission spectra of 375 nm LED (**a**), 400 nm LED (**b**), and 455 nm LED (**c**). Emission spectra of Rb_2_Eu(PO_4_)(MoO_4_) ceramic disks excited at 375 nm LED (**d**), 400 nm LED (**e**), and 455 nm LED (**f**).

**Figure 8 materials-12-03275-f008:**
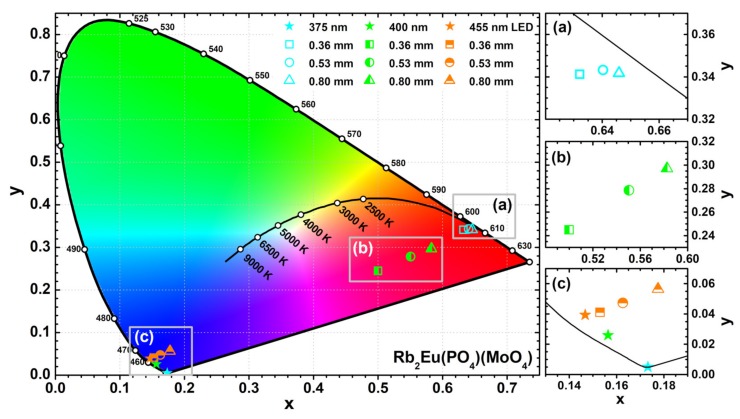
CIE 1931 colour space diagram with the colour coordinates of 0.36, 0.53, and 0.80 mm Rb_2_Eu(PO_4_)(MoO_4_) ceramic disks combined with 375, 400, and 455 nm emitting LEDs. Magnified parts show enlarged areas of CIE 1931 diagram with colour coordinates of 0.36, 0.53, and 0.80 mm Rb_2_Eu(PO_4_)(MoO_4_) ceramic disks excited with: 375 nm LED (**a**), 400 nm LED (**b**), 455 nm LED (**c**).

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
