# Peer review of "Optical Properties of Red-Emitting Rb2Bi(PO4)(MoO4):Eu3+ Powders and Ceramics with High Quantum Efficiency for White LEDs"

_materials, 2019, doi:10.3390/ma12193275_

Round 1

Reviewer 1 Report

In the paper "Optical Properties of Red Emitting Rb2Bi(PO4):Eu3+ Powders and Ceramics with High Quantum Efficiency for White LEDs" have presented in well written way the results of the study.

However there are some small inconsistencies in the manuscript which after revision will make this paper even better presented. The authors are encouraged to check the labeling of the Figures and Tables in the manuscript as there are sometimes written in the bold and italics and some are written as the rest of the text.

The sentence on the page 8. line 268 ... 100% Eu3+ doped samples are depicted in Figure 5a, Figure 5b, Figure 5c, and Figure 5d, respectively. It would sound better if authors would write it ... 100% Eu3+ doped samples are presented in Figure 5a-d respectively. Also the listing of Figure 7a, Figure 7b, and Figure 7c, respectively. on page 9. line 320 and 321 instead should be written Figure 7a-c repectively.

The Figure 8. the labels of the 375 nm in the legend and also in the CIE diagram are not visible as the color that was used is a little bit not enough in the contrast with red region and also the labels are overlapped. I would suggest to authors to use darker color for labels. Also it would be useful to make an inlet where it would be magnified region where are these labels.

1. The authors used for photoluminescence decay kinetics μ-flash lamp with pulse repetition of 25 Hz. Why have you not used the repetition of 100 Hz?

Reviewer 2 Report

The work of Grigorjevaite et al concerns the optical properties of Eu3+ doped Rb2Bi(PO4)(MoO4) materials obtained by a solid state method. In my opinion a minor revision is needed due to the following remarks and questions:

Page 3: I think the QE expressed in page 3 and depicted in Fig.6 corresponds to an internal QE, i.e. the ratio between emitted photons and absorbed photons as given in expression (1). Did the authors make a confusion with the external QE ?

Please give the cell parameters of Rb2Bi(PO4)(MoO4) to prove the isostructurality with the potassium compound.

The crystal structure of Rb2Bi(PO4)(MoO4) was already published by Xia et al (Solid State Com. 201 (2015) 102-106). Please add this reference.

Line 160: The 7F0-5FJ transition was already given in line 159.

Bi3+ cation is known to be optically active. Is there any contribution of this ion for instance in excitation spectra?

Line 174: the 5D0-7F0 component can be observed in emission spectra since the local site is C2.

Page 6 and Table S2: why lifetimes are greater for a 265 nm excitation?

Line 224: 5D1 is written twice.

Table S3 and Fig. 4d: the temperature dependent lifetime behaviour for the concentrated sample is unusual (maximum at 400K). Please check your values or record them again.

Fig.6: it is surprising to observe that the QE is lower for a 265 nm excitation while the intensity of the CTS band in excitation spectra is relatively high. Is there any explanation?

The table S6 is wrong (probably the first two columns are not correct).

Reviewer 3 Report

This paper on an Eu3+ doped phosphor is well-written and merits publication in an international journal. However, a few issues have to be resolved before the paper is suitable for publication:

While the level of English is quite good in general, there are numerous smallish language errors which can be corrected upon careful proofreading. In the introduction (line 54) the authors correctly state that phosphors with a low production cost are needed. However, the optimum phosphor, as described in the paper, has a 50% Eu-concentration, which will make the phosphor extremely expensive. A critical note on phosphor cost might be in place in the conclusions. In the introduction, Mn4+ could be mentioned as an alternative dopant for narrowband red emission. While it is an original way to show a reference XRD pattern, I would suggest the authors to adopt the more conservative approach in figure 1, and flip the reference pattern upside down. In the inset of figure 3b, vertical axis numbers are missing. Please add the excitation wavelength (393.5 nm ?) to the caption for the inset of figure 3b. Please add a unit (%) to the horizontal axis title of figure 6. Line 307: “… excitation at 265 and 465 has …”: units (nm) missing. Line 307: “… increases the path …”: this is confusing, because actually, since the scattering is suppressed in a ceramic, the path length is reduced, whereby the light can reach deeper in the phosphor and thus excite more dopants. Please check the phrasing of the statement. The fact that the excitation lines are very narrow (nicely illustrated in figure 7) is a marked disadvantage of this type of phosphor: for any LED excitation, the external quantum efficiency is necessarily very low. The authors are suggested to add a critical note on this. (On the positive side, they could claim that a major part of the exciting LED emission remains available for exciting phosphors for the other colors). The 716 K quenching temperature in the conclusions is, without additional information, misleading since this is based on a fit/extrapolation of the experiments. While not wrong, I would suggest the authors to remove the columns with the 1976 color coordinates in the supporting information, since this adds a lot of unnecessary information to the tables. Any interested reader can easily convert the (x,y) coordinates if necessary.

Round 2

Reviewer 2 Report

The article of Grigorjevaite et al can be accepted for publication. All corrections have been taken into account.